# pH-Responsive Hybrid Nanoassemblies for Cancer Treatment: Formulation Development, Optimization, and In Vitro Therapeutic Performance

**DOI:** 10.3390/pharmaceutics15020326

**Published:** 2023-01-18

**Authors:** Patrícia V. Teixeira, Filomena Adega, Paula Martins-Lopes, Raul Machado, Carla M. Lopes, Marlene Lúcio

**Affiliations:** 1CF-UM-UP—Centro de Física das Universidades do Minho e Porto, Departamento de Física, Universidade do Minho, 4710-057 Braga, Portugal; 2DNA & RNA Sensing Lab, Department of Genetics and Biotechnology, University of Trás-os-Montes and Alto Douro, Blocos Laboratoriais Ed, 5000-801 Vila Real, Portugal; 3CAG—Laboratory of Cytogenomics and Animal Genomics, Department of Genetics and Biotechnology, University of Trás-os-Montes e Alto Douro, 5000-801 Vila Real, Portugal; 4BioISI—Biosystems and Integrative Sciences Institute, Faculty of Sciences, University of Lisbon, 1749-016 Lisboa, Portugal; 5CBMA—Center of Molecular and Environmental Biology, Departamento de Biologia, Universidade do Minho, 4710-057 Braga, Portugal; 6IB-S—Institute of Science and Innovation for Bio-Sustainability, Campus de Gualtar, University of Minho, 4710-057 Braga, Portugal; 7Instituto de Investigação, Inovação e Desenvolvimento (FP-I3ID), Biomedical and Health Sciences Research Unit (FP-BHS), Faculdade de Ciências da Saúde, Universidade Fernando Pessoa, Rua Carlos da Maia 296, 4200-150 Porto, Portugal

**Keywords:** cancer, chemotherapeutic agents, doxorubicin, drug-delivery systems, lyotropic nonlamellar liquid crystalline nanoassemblies

## Abstract

Current needs for increased drug delivery carrier efficacy and specificity in cancer necessitate the adoption of intelligent materials that respond to environmental stimuli. Therefore, we developed and optimized pH-triggered drug delivery nanoassemblies that exhibit an increased release of doxorubicin (DOX) in acidic conditions typical of cancer tissues and endosomal vesicles (pH 5.5) while exhibiting significantly lower release under normal physiological conditions (pH 7.5), indicating the potential to reduce cytotoxicity in healthy cells. The hybrid (polymeric/lipid) composition of the lyotropic non-lamellar liquid crystalline (LNLCs) nanoassemblies demonstrated high encapsulation efficiency of the drug (>90%) and high drug loading content (>7%) with colloidal stability lasting at least 4 weeks. Confocal microscopy revealed cancer cellular uptake and DOX-loaded LNLCs accumulation near the nucleus of human hepatocellular carcinoma cells, with a large number of cells appearing to be in apoptosis. DOX-loaded LNLCs have also shown higher citotoxicity in cancer cell lines (MDA-MB 231 and HepG2 cell lines after 24 h and in NCI-H1299 cell line after 48 h) when compared to free drug. After 24 h, free DOX was found to have higher cytotoxicity than DOX-loaded LNLCs and empty LNLCs in the normal cell line. Overall, the results demonstrate that DOX-loaded LNLCs have the potential to be explored in cancer therapy.

## 1. Introduction

Cancer is one of the leading causes of death globally [1]. In 2020, the International Agency for Research on Cancer reported 19.3 million new cancer cases and approximately 10 million cancer deaths [2]. Furthermore, cancer incidence and mortality are rising worldwide, with 28.4 million new cases expected by 2040.

Cancer treatment is extremely complex because each tumour cell presents a unique genetic profile and may be resistant to available chemotherapeutic agents [3]. Doxorubicin (DOX) is a common chemotherapeutic agent used to treat cancers such as carcinomas, sarcomas, and hematologic neoplasms [4]. DOX’s mechanism of action is quite complex, involving DNA intercalation that inhibits a key enzyme, topoisomerase II, as well as the production of reactive oxygen species (ROS) [5]. Despite its high efficacy, DOX has some drawbacks in clinical use, including poor specificity and dose-dependent irreversible cardiotoxicity [6,7]. As a result, drug delivery nanosystems that control DOX release and uptake at the tumour site while minimizing toxicity to healthy cells are required [8,9].

In recent decades, lyotropic non-lamellar liquid crystalline (LNLCs) nanoassemblies, formed by amphiphilic lipids and surfactants (lipids or polymers) mixed in a specific ratio, have received increased attention in both scientific and applied research [10,11,12,13]. The internal structures of LNLCs are made up of inverse bicontinuous cubic mesophases, which can accommodate a wide range of therapeutic agents, from small drug molecules to biomacromolecules that are both hydrophilic and hydrophobic, or even amphiphilic [14]. Because of their large internal surface area, LNLCs achieve high loading contents of chemotherapeutic agents [15,16,17,18,19] as well as the combined delivery of different active agents [20,21,22,23,24]. Furthermore, the existence of an internal structure of bicontinuous water and oil channels allows drugs to be located differently depending on their ionization. In this regard, we developed pH-triggered LNLCs composed of a lipid, glyceryl monooleate (GMO), and a polymer, poloxamer P407 (P407), loaded with DOX in order to improve its anticancer effect, minimizing the cytotoxic effect in healthy cells. At the low-pH environment of cancer tissues and endosomal cancer cell vesicles (pH 5.5), DOX is fully protonated and thus located in the aqueous channels of the LNLC nanoassemblies with a higher release rate. However, at higher physiological pH (pH 7.4), typical of healthy cells, the protonated fraction of the drug decreases while the DOX zwitterionic fraction increases, making it more lipophilic and preferentially located in the lipidic domains, resulting in significantly delayed drug release [25,26] (Figure 1).

DOX-loaded LNLCs have been the focus of other studies [20,21,25,26,27,28] due to the great potential of the mesophases to control the delivery of the drugs carried. However, our study proposes a rational development of the LNLCs methods of preparation and loading regarding colloidal stability which the other authors have failed to take into account. The storage colloidal stability of the LNLCs involved a comprehensive physicochemical characterization and was evaluated to determine if the mean particle size, PDI, surface charge and drug loading changed over time, as this could be a sign of membrane rupture, aggregation, or sedimentation of the nanoassemblies or a sign of loss therapeutic efficiency due to early drug release, and thus can have a strong impact in biological performance [29]. Furthermore, to achieve safer formulations, a different proposed lipid:polymer ratio was used, favoring the reduction in polymeric coating, since it has been reported that P407, although generally accepted as safe, is non-biodegradable and in higher doses has demonstrated cellular toxicity [30]. Finally, in addition to assessing the cytotoxic effect against breast cancer cells and fibroblasts (normal cells highly affected by DOX toxicity), the cellular uptake and cytotoxic effect of DOX-loaded LNLCs in hepatic and lung cancer cells was investigated for the first time.

## 2. Materials and Methods

### 2.1. Materials

Doxorubicin hydrochloride (DOX), with purity ≥98%, was purchased from Sigma Aldrich (St. Louis, MO, USA). Glyceryl monooleate (GMO; Peceol^®^) was kindly provided by Gattefossé (Lyon, France). Pluronic^TM^ F-127, also known as poloxamer P407 (P407), was obtained from Merck KGaA (Darmstadt, Germany). Dulbecco’s Modified Eagle’s Medium (DMEM), AmnioMax, supplement, fetal bovine serum (FBS), L-glutamine, antibiotics (penicillin G and streptomycin), and sulforhodamine B (SRB) were purchased from Thermo Fisher Scientific (Waltham, MA, USA). 

All other reagents were acquired from Merck KGaA (Darmstadt, Germany) with p.a. quality and used without further purification. All aqueous solutions were prepared with ultrapure water produced by Millipore Sigma (Burlington, MA, USA) Milli-Q^®^ system (resistivity = 18.2 MΩ·cm).

### 2.2. Methods

#### 2.2.1. LNLCs Preparation Method

Empty-LNLCs were prepared using three different methods: hydrotrope, lipid film hydration, and emulsification.

The hydrotrope-based method was adapted from a study performed by Abdelaziz et al. [31]. Briefly, an isotropic mixture of GMO (4% *w/v*) and absolute ethanol (450 μL) was prepared and dropwise added into 5 mL of aqueous polymeric solution of poloxamer P407 (0.5% *w/v*). The mixture was then kept at room temperature (25 °C) for 24 h. Following that, the remaining 15 mL of polymeric solution was added to the mixture under stirring. Finally, the mixture was homogenized (Ultraturrax, Polytron^®^ PT 2500, VWR International, Radnor, PA, USA) for 3 min at 13,500 rpm to obtain the final dispersion.

The lipid film hydration method was adapted from a study conducted by Freag et al. [32]. In brief, GMO (4% *w/v*) and P407 (5% *w/v*) were dissolved in absolute ethanol and evaporated to dryness for 2 h in a rotary evaporator (IKA, Staufen, Germany). The lipid film was then hydrated with 20 mL of ultrapure water heated above the GMO phase transition temperature (45 °C). After, the mixture was homogenized (Ultraturrax, Polytron^®^ PT 2500, VWR International, Radnor, PA, USA) for 3 min at 13,500 rpm, and then sonicated for 3 min at 18 W (Misonix S-4000 sonicator, Misonix™, Farmingdale, NY, USA). The final dispersion was obtained by extruding the sample (Lipex extruder, Northern Lipids Inc., Burnaby, BC V5J 5G7, Canada) at 6–8 bar pressure through a 400 nm polycarbonate filter (Isopore^TM^ membrane filter, Millipore Sigma (Burlington, MA, USA).

The emulsification method was adapted from a study performed by Freag et al. [32]. In summary, lipid and polymer (GMO, 4% *w/v*, and P407, 5% *w/v*) were mixed and heated in a water bath at 70 °C to dissolve the polymer. This mixture was dropped into ultrapure water, which was preheated to 70 °C for 15 min with magnetic stirring (IKA^®^C- MAG HS 4, IKA^®^-Werke GmbH & Co. KG, Staufen, Germany). The final dispersion was homogenized (Ultraturrax, Polytron^®^ PT 2500, VWR International, Radnor, PA, USA) for 3 min at 13,500 rpm, and then sonicated for 3 min (Misonix S-4000 sonicator, Misonix™, Farmingdale, NY, USA) at 18 W.

To select the best LNLC preparation method, the colloidal stability of LNLCs obtained by each method was evaluated in terms of mean particle size and polydispersity index (PDI) by dynamic light scattering (DLS), and zeta potential (ZP) by electrophoretic light scattering (ELS) using the Anton Paar Litesizer^®^ 500 (Anton Paar GmbH, Graz, Austria) over a 4-week period at refrigerated conditions (4 °C) and room temperature (25 °C).

#### 2.2.2. DOX Loading Method

For each LNLCs preparation method selected, three different procedures were used to prepare DOX-loaded LNLCs (DOX-LNLCs): direct mixing, hydration, and incubation. DOX (0.03 mg/mL) was added directly to the lipid phase in the direct mixing procedure, or to the aqueous phase in the hydration procedure, and the LNLCs preparation process followed the same steps as previously described. In the incubation procedure, DOX (0.03 mg/mL) was dissolved in absolute ethanol and evaporated to dryness for 2 h in a rotary evaporator (IKA, Staufen, Germany) and then incubated with the previously prepared LNLCs at a temperature above the melting point (Tm) of GMO for one hour. To select the best DOX-loading method, the colloidal stability of DOX-LNLCs was evaluated as previously described. Then, different DOX concentrations (0.03 mg/mL, 0.33 mg/mL, 0.98 mg/mL, and 3.26 mg/mL) were loaded in LNLCs to determine whether the produced nanoassemblies have a maximum loading capacity and the ways in which the increase in concentration affects the physicochemical characteristics of the LNLCs.

#### 2.2.3. LNLCs Physicochemical Characterization

The final selected empty LNLCs and DOX-LNLCs were characetreized for mean particle size, PDI, and ZP by DLS and ELS using the Anton Paar Litesizer^®^ 500 (Anton Paar GmbH, Graz, Austria). To avoid multiple scattering effect due to the high particle concentration, the formulation was previously diluted with ultrapure water and placed in Omega cuvette Mat. No. 225288 (Anton Paar GmbH, Graz, Austria) at 25 ± 1 °C. Size and PDI results were obtained from the correlogram using Anton Paar Litesizer^®^ 500 software (Anton Paar GmbH, Graz, Austria) after cumulant analysis according to ISO 22412:2008 [33] and ZP results were obtained via the conversion of electrophoretic mobility according to the method used by Helmholtz–von Smoluchowski [34].

The shape and morphology of the LNLCs was examined by scanning transmission electron microscopy, STEM (Hitachi HD-2700 scanning transmission electron microscope, Tokyo, Japan). First, 10 µL of LNLCs was added to carbon grids and left standing for 2 min. The liquid in excess was removed with filter paper, and afterwards, 10 µL of 4% neodymium acetate was added to the grids for 10 s for negative staining. Visualization was carried out on a Hitachi HD-2700 scanning transmission electron microscope at 200 kV. 

To evaluate the chemical composition of the LNLCs, Fourier transform infrared spectroscopy (FTIR) analysis was performed using a FTIR spectrophotometer (Spectrum Two™ FTIR Perkin-Elmer, Waltham, MA, USA) equipped with an attenuated total reflectance unit (ATR-FTIR). Samples (free DOX, empty LNLCs and DOX-LNLCs) were prepared by adding 200 μL of DOX aqueous solution (0.98 mg/mL) or LLCNs into aluminium crucibles, and water was evaporated at room temperature to form a film. Measurements were performed by pressing the film against the crystal of the ATR accessory in the range between 400 and 4000 cm^−1^, with a resolution of 4 cm^−1^ and accumulating 64 scans per spectrum.

The phase transition temperature of empty LNLCs and DOX-LNLCs was performed by differential scanning calorimetry (DSC). Briefly, 10 to 13 mg of the LNLCs samples were placed into sealed aluminium crucibles with perforated caps. The thermal analysis profiles were obtained under a nitrogen dynamic atmosphere (20 mL/min). The thermal program involved cooling down to 5 °C (at a rate of 10 °C/min), followed by an isotherm of 8 min and subsequent heating from 30 to 70 °C (at a rate of 5 °C/min). Thermograms of free DOX, empty LNLCs and DOX-LNLCs were recorded using DSC 200 F3 Maia^®^ instrument (NETZCH, Bobingen, Germany).

The thermodynamic behaviour of LNLCs was also studied by DLS. In brief, 1 mL of each sample was placed in a disposable polyester cell, and the size was measured using the DLS technique as previously described. The intensity and average count rate of light intensity scattered by the LNLCs were measured at temperatures ranging from 30 °C to 70 °C, with a 1 °C difference between measurements, and three measurements were taken at each temperature.

Encapsulation efficiency (EE) was determined by ultrafiltration technique [35]. In brief, DOX-LNLCs previously diluted in water (1:2, *v/v*) were transferred to filter units with 50 kDa pores (Amicon^®^ Ultra-15, Millipore Corporation, Burlington, MA, USA) and centrifuged at 3000 rpm for 10 min (Hettich^®^ Universal 320 centrifuge, Kirchlengern, Germany). The encapsulated DOX was quantified by an International Conference on Harmonization (ICH)-validated UV/Vis spectrophotometry method using a Shimadzu UV/Vis-NIR 3101 PC spectrophotometer (Kyoto, Japan) at the maximum absorption wavelength of DOX, which was estimated to be 478 nm in agreement with previous reports [36,37].

EE and drug loading (DL) content were calculated indirectly by applying the following Equations:(1)$$\mathrm{EE}\;(\%)=\frac{\left[\mathrm{DOX}\right]\mathrm{Initial}-\left[\mathrm{DOX}\right]\mathrm{Free}}{\left[\mathrm{DOX}\right]\mathrm{Initial}}\times 100$$
(2)$$\mathrm{DL}\;(\%)=\frac{\left(\left[\mathrm{DOX}\right]\mathrm{Initial}\;\times \;\mathrm{EE}\;\left(\%\right)\right)}{\left[\mathrm{LNLCs}\right]}\times 100$$
where [DOX]_Free_ represents the concentration (M) of DOX present in the supernatant, [Drug]_Initial_ refers to the concentration (M) that was firstly added, and [LNLCs] refers to the formulation concentration (M). In all cases, DOX solubility in water was assured as the concentrations used never reached its maximum solubility (50.0–52.0 mg/mL according to supplier information).

#### 2.2.4. Potential Therapeutic Performance

In vitro drug release studies were performed by the dialysis method [35]. DOX-LNLCs (1.0 mL) with DOX concentration of 0.03 mg/mL were added to dialysis membranes (Float-A-Lyzer^®^, 3.5 kD, VWR International, Radnor, PA, USA) that were immersed in 40 mL of dissolution medium, ensuring sink conditions. The dissolution medium composed of buffer solutions used to mimic the pH values found in vivo: pH of 5.5 mimicking more acidic conditions found in tumour microenvironment tissues, specifically at endosomal vesicles and pH of 7.4 mimicking the physiological environment of normal tissues or plasma [38]. This system was kept at 37 ± 1 °C under orbital stirring (IKA^®^ KS 3000 i control, Staufen, Germany), at 250 rpm, to also mimic the body temperature. At predefined intervals, 1.0 mL of each sample was collected and replaced with 1.0 mL of fresh dissolution medium until the assay ran for 25 h. The amount of DOX released in the dissolution medium was determined using a validated fluorescence spectrophotometric method at the maximum emission wavelength of DOX, which was estimated to be 595 nm, in agreement with the previously reported value [37]. The cumulative DOX released was expressed as a percentage of the theoretical maximum drug content value.

Human dermal fibroblast neonatal (HDFn), human hepatocellular carcinoma (HepG2), human non-small cell lung cancer (NCI-H1299) and human breast metastatic adenocarcinoma (MDA-MB-231) cell lines were acquired from the American Type Culture Collection (ATCC, USA). Cells were maintained and grown at 37 °C under 5% CO_2_ in DMEM enriched with 10% (*v/v*) FBS, 13% (*v/v*) AmnioMax, 0.5% (*v/v*) supplement, 1% (*v/v*) L-glutamine and antibiotics (10,000 U/mL penicillin G and 100 mg/mL streptomycin).

To evaluate growth inhibition (i.e., the concentration that causes 50% inhibition of cell proliferation—GI_50_)/cytotoxicity of the formulations, a previously described sulforhodamine B assay method (SRB) was used [39]. Briefly, cell lines were seeded in 96-well flat-bottomed polystyrene multiwell plates at a density of 3 × 10^4^ cells/well and incubated under humidified 5% CO_2_ atmosphere at 37 °C for approximately 2 to 3 h to promote conditions for cell adhesion. After seeding, the medium was removed and the attached cells were incubated for 24 and 48 h with different concentrations of free DOX, empty LNLCs, and DOX-LNLCs, at a range of DOX concentrations from 180 to 0.05 μM. Untreated cells constituted the negative control. After the treatment, the cells were fixed with 10% (*w/v*) trichloroacetic acid (TCA) cold solution at 4 °C for 1 h, then they were washed with slow-running tap water and dried. Afterwards, when the cells completely dried, a staining step with SRB solution (0.057% (*w/v*) SRB in 1% (*v/v*) acetic acid) for 30 min at room temperature was performed. The plates were then washed with 1% (*v/v*) acetic acid solution and dried thoroughly. The protein-bound dye was finally solubilized in a 10 mM Tris base solution with a pH of 10.5 and shaken on a rotary shaker until complete solubilization of SRB. Finally, the optical density (OD) was measured in a microplate reader at 510 nm. Dose–response curves were obtained for each tested compound and cell line, and the GI_50_ values were calculated as described by Vichai and Kirtikara [39] and expressed as mean ± standard deviation (SD) of three independent experiments.

In the uptake cellular studies, HepG2 cells were seeded in glass slides at a density of 3 × 10^4^ and incubated under humidified 5% CO_2_ atmosphere at 37 °C for approximately 5 h to promote cell adhesion. After seeding, the medium was removed, and the attached cells were incubated for 24 h with DOX-LNLCs at DOX concentrations of 10, 5 and 0.05 μM. Untreated cells constituted the control. After treatment, the glass slides were double washed with phosphate-buffered saline solution (PBS 1x) at 37 °C, fixed with paraformaldehyde solution in PBS 1x (2% *w/v*) for 20 min at room temperature, and then washed again with PBS 1x at room temperature three times, to completely remove any paraformaldehyde remnants. Finally, the glass slides were mounted with medium containing 4′-6-diamidino-2-phenylindole (DAPI) (Vector Laboratories, Newark, CA, USA) and stored in the dark at 4 °C until observation. Following that, confocal images of the cells were acquired on an LSM 510 META with a Zeiss Axio Imager Z1 microscope (Carl Zeiss, Oberkochen, Germany) and LSM 510 software (version 4.0 SP2) (Carl Zeiss, Oberkochen, Germany). The same microscope settings were applied to all images to allow results normalization. The lasers used were helium–neon (543 nm) set at approximately 50% and diode (405 nm) set at approximately 69%. The pinhole was set to 102 mm (0.98 airy units) for helium–neon laser, and 112 mm for diode laser using a 63× objective. Images were captured at a scan speed of 6 with 1 µm-thick Z sections, deconvolutioned using the 3D deconvolution tool of the AutoQuant X3 software (Media Cybernetics, Rockville, MD, USA) and processed in TIFF images with ImageJ (1.47v) (National Institutes of Health, Bethesda, MD, USA).

#### 2.2.5. Statistical Analysis

All data are represented as the mean ± standard deviation (SD) of three independent assays. Data were analysed with the ANOVA test followed by the Bonferroni post hoc pairwise comparison procedure used to determine whether changes were statistically significant (*p*-value < 0.05). The GI_50_ sigmoid curves were obtained using GraphPad^®^ Prism^®^ software version 8.0 (Boston, MA, USA) and GI_50_ values and standard deviation were calculated using non-linear regression analysis.

## 3. Results and Discussion

### 3.1. Selection of the LNLCs Preparation Method

The empty LNLCs were prepared using three different methods, one by the bottom-up approach (hydrotrope method (M1)) and the other two by the top-down approach (lipid film hydration method (M2) and emulsification method (M3)). The physicochemical properties (particle sizes, ZP and PDI) of the empty LNLCs in the moment of preparation (Figure 2) and the stability of those properties during storage (Figure 3) were evaluated to select the best preparation method(s). 

The mean particle size and PDI of empty LNLCs M1 (233 ± 53 nm; 0.25 ± 0.04), empty LNLCs M2 (208 ± 32 nm; 0.19 ± 0.02)) and empty LNLCs M3 (241 ± 35 nm; 0.23 ± 0.013) were statistically similar and were not significantly different from those obtained by other authors [14,18,19,40,41,42,43,44,45,46]. All three methods proved to be effective in producing LNLCs with an acceptable PDI value for these types of nanoassemblies [47], confirming the formulation’s homogeneity. 

In addition, the LNLCs surface charge was determined by measuring their ZP values (Figure 2B). Although the nanoassemblies are coated with a non-ionic polymer (P407), the ZP values of empty LNLCs M1 (−17.6 ± 7.9 mV), M2 (−13.9 ± 1.94 mV) and M3 (−18.4 ± 4.2 mV) indicate a negatively charged surface, in agreement with other reports using non-ionic coatings [40,48,49], possibly due to some free fatty acids resultant from lipid or polymer hydrolysis. 

The colloidal stability of the LNLCs was evaluated to determine the physicochemical properties under different storage conditions. Figure 3 depicts mean particle size, PDI, and ZP values of empty LNLCs prepared using various methods and stored at different temperatures (4 °C vs. 25 °C).

Overall, the maintenance of physicochemical properties in long term-storage confirmed the colloidal stability of any of the prepared empty LNLCs, indicating a lack of aggregation during the time period evaluated and at the storage conditions tested. Therefore, contrastingly to other authors that indicate top-down methods as producing more stable cubosomes dispersions [50], we did not detect significant differences between the methods of LNLCs production, and we believe that the differences found in the literature may be more dependent on the ratio of lipid:polymer than on the method of production used. Indeed, the polymeric surfactant P407 has an important role in the colloidal stability as it sterically stabilizes the interface by adsorbing and maintaining its hydrophobic poly(propylene oxide) (PPO) blocks on the surface of the nanoassemblies, thereby maintaining the internal structure of the inverted non-lamellar mesophases [50]. Furthermore, the steric effect conferred by polymeric coating compensates the relatively large sizes of the LNLCs. Indeed, such polymeric coating is described as a means of avoiding detection by the immune system’s macrophages and phagocytic system due to minimal protein binding on the surface of the nanoassemblies, which may be also beneficial for in vivo stability [51,52], maximizing formulation circulation time, as well as for taking advantage of the enhanced permeability and retention (EPR) effect [53]. 

Although no significant differences in either storage condition or production methods were observed, LNLCs prepared using the top-down approach (i.e., lipid film hydration (M2) and emulsification (M3) methods) and the storage condition at 25 °C were chosen due to smaller oscillations (i.e., smaller error bars) in mean particle size, PDI and ZP over time when compared to M1 and storage at 4 °C.

### 3.2. Selection of the DOX Loading Method

Following the selection of LNLCs M2 and M3, we evaluated three DOX loading methods to determine the more efficient one. The data in terms of mean particle size, PDI, and ZP values of LNLCs prepared with and without DOX stored for 4 weeks at 25 °C are presented in Figure 4.

After 4 weeks, there were no significant changes, indicating a lack of clustering which is supported by a PDI value less than 0.25, an acceptable value that is consistent with a homogeneous population of nanoassemblies.

Overall, the changes in size, PDI, and surface charge over time are not statistically significant, which may lead to the conclusion that these formulations (DOX-M2 and DOX-M3) for the three encapsulation methods (incubation, hydration, and direct mixing) are stable. However, macroscopic color changes in the formulations were observed after 4 weeks compared to measurements taken immediately after formulation preparation (Appendix A) indicating that DOX may have been located at the lipid/water interface of the LNLCs over time.

The DOX loading procedures were tested to determine the most effective one. Appendix A depicts the encapsulation efficiency (EE%) and drug loading (DL%) of DOX-LNLCs M2 and M3 using different loading methods.

The direct mixing method was the encapsulation method chosen as it proved to be the most efficient, with significantly higher encapsulation efficiency (EE%) and drug loading (DL%) (*p*-value < 0.05) for DOX-LNLCs M2 (84.9 ± 6.6% and 0.06 ± 0.005%, respectively) and DOX-LNLCs M3 (89.9 ± 6.2% and 0.06 ± 0.004%, respectively) compared to the hydration and incubation methods (Appendix A). The differences between the methods are explained by the way that DOX is encapsulated in the LNLCs, as each method offers a different structural organization.

### 3.3. LNLCs Physicochemical Characterization

The mean size, PDI, and ZP values were used to evaluate the LNLCs produced by the chosen method of preparation (M3) and loaded with different DOX concentrations by the chosen loading method (direct mixing) (Figure 5).

The LNLCs with DOX at 0.98 mg/mL (165.8 ± 1.7 nm) showed a statistically significant size reduction (*p*-value < 0.05) when compared to the empty LNLCs (240.6 ± 35.3 nm) (Figure 5A), which can be explained by the effect of DOX concentration in lipid packing causing some lipid restruturing and size changes. The PDI did not differ significantly from empty LNLCs, remaining approximately 0.25 for all formulations, indicating an homogenous dispersion. These physicochemical properties were maintained over time, confirming their colloidal stability.

At the highest DOX concentrations (0.33, 0.98, and 3.26 mg/L), the ZP value increased significantly (−1.03 ± 0.95, +0.28 ± 0.28, and +1.42 ± 0.98 mV, respectively, *p*-value < 0.05) (Figure 5B) which can be explained by the positively charged drug screening the negative surface charge at the lipid/water interface.

Figure 6A shows the characteristic LNLCs structure, with the facets of the nanoassemblies consistent with crystallographic planes of a cubic and/or hexagonal morphology and a hydrodynamic diameter of less than 250 nm (Figure 6B), confirming the data obtained by DLS (Figure 5A).

The biophysical and thermodynamic properties of LNLCs are consistent with the crystallographic planes observed (Appendix A). The thermodynamic characterisation of DOX-LNLCs M3 by DLS and corroborated by DSC reveals a phase transition, which may correspond to the transition between the P-type cubic phase and the D-type cubic phase (Appendix A).

EE was greater than 90% for the higher DOX concentration tested (Figure 7), which was comparable to the findings of Godlewska et al. (93 ± 4%), Nazaruk et al. (92 ± 4%) and Muheem et al. (92.3 ± 4.99%) [25,42,54]. DOX concentration increment resulted in a significant increase in DL (*p*-value < 0.05), up to 7.12% (Figure 7). After 4 weeks, the EE and DL values for DOX-M3 (0.98 mg/mL) formulation were 84.63 ± 4.7% and 1.84 ± 0.102%, respectively, indicating no significant changes over time.

The chemical characterization of hybrid nanoassemblies was evaluated using ATR-FTIR (Figure 8).

The ATR-FTIR spectra of empty LNLCs M3 (Figure 8A) present (CH_3_) and (CH_2_) bending vibrations at 1375 and 1470 cm^−1^, and Vs(CH_2_) and Vas(CH_2_) stretching vibrations at 2860 and 2930 cm^−1^ (band C) characteristic of aliphatic chains [55,56] present in both polymer (P407) and lipid (GMO). The asymmetric stretching vibrations of the unsaturation (trans) C=C assigned at 962 and 1670 cm^−1^ are only due to GMO component of LNLCs (Figure 8B) [55,57]. In addition, the stretching vibration of the carbonyl group (C=O) from the ester bond at 1670 cm^−1^ can only be attributed to GMO component of LNLCs. Both empty LNLCs and DOX-LNLC contain -C-O groups, but their assignment is slightly different permitting to identify the drug component (Figure 8B, Vs(C-O)). P407 has several ether groups that appear at lower frequencies (1100 cm^−1^), whereas in the case of DOX, the -C-O group is part of the ester bond (O-C=O) and thus appears more intense at higher frequencies (Figure 8B, Vs(C-O)) [55,58]. At roughly the same frequencies and probably superimposed are the stretching modes of the hydroxyl groups from GMO (–C–OH β at ≈1116 cm^−1^ and –C–OH γ at ≈1150 cm^−1^) (Figure 8B). The broad absorption band at 3200–3400 cm^−1^ (band D) is also attributed to the O–H stretching mode of the water, GMO and P407 [55,57]. Another slightly different aspect between the spectra of empty LNLCs and DOX-LNLC is located at the 808–878 cm^−1^ and might be attributed to C=H bending vibration also present in DOX spectrum (Figure 8B, δ(C=H)) as reported [59,60].

### 3.4. Potential Therapeutic Performance

Several parameters can be used to predict formulation therapeutic potential in vitro. Firstly, the DOX release profile from LNLCs was evaluated at different pH values, as it is a critical feature for the formulation’s in vivo performance (Figure 9). Compared to normal tissues, which have a pH range of 7.4–7.5, tumour tissues have an acidic microenvironment (pH 6.0 to 7.0) due to hypoxia and extensive cell death, and this acidity increases (pH 5.0 to 6.0) upon intracellular delivery by endocytic vesicles [52,61,62].

At pH 5.5, DOX was released from LNLCs M3, up to a maximum of 23.3 ± 2.4% with a first-order kinetic constant rate of 0.63 ± 0.24 h^−1^. However, at pH 7.4, significantly less DOX was released (8.5 ± 0.8%, *p*-value < 0.05) with a a first order kinetic constant rate of 0.88 ± 0.33 h^−1^ (Figure 9). The non-linear drug release profile was fitted with other mathematical models (Weibull and Gallagher–Corrigan, shown in Appendix A). In agreement with other reported studies [20,25,26,27,54], DOX-LNLCs higher drug release at acidic pH may be explained by DOX ionization properties. In the low-pH environment of cancer tissues (pH ≈ 6.0) and at pH of endocytic vesicles (pH 5.5), the drug is almost completely protonated (99.8 %) and is located in the aqueous channels of the LNLC nanoassemblies with a higher release rate than in the lipid domain. However, at higher physiological pH (pH 7.4), the DOX protonated fraction decreases to 38.7 %, with the remainder in the zwitterionic state. The drug’s electroneutrality makes it more lipophilic, and thus its location is preferentially in the lipidic domains, resulting in significantly delayed drug release (*p*-value < 0.05) under these conditions. Overall, accelerated DOX release in a simulated tumour environment and significantly smaller release (*p*-value < 0.05) under normal physiological conditions were observed, indicating that these formulations have a pH-triggering strategy with potential to reduce cytotoxicity in healthy cells.

By measuring cell growth, the in vitro cytotoxicity of empty LNLCs and DOX-LNLCs was evaluated in different tumor cell lines (MDA-MB-231, human breast adenocarcinoma; NCI-H1299, human non-small cell lung cancer; HepG2, human hepatocellular carcinoma); and in comparison to free DOX. In addition, a normal cell line (HDFn) was used to evaluate the cytotoxicity of the nanoassemblies in off-target tissues (Appendix A). The cytotoxicity was evaluated using Generally Recognized As Safe (GRAS) standards, which classify in vitro as potentially cytotoxic if a reduction in cell viability by 30% or more [63] is observed.

The empty LNLCs generally showed minimal cytotoxic activity at lower concentrations (≤5.0 μM); however, at higher concentrations, cytotoxic activity increased, and cell growth decreased more abruptly (Appendix A). This suggests that empty LNLCs are biocompatible at selected doses. DOX-LNLCs resulted in a greater decrease in cell growth (*p*-value < 0.05) than the free DOX at 48 h, at concentrations of 20 and 45 μM, in all tumor cell lines tested, and at 24 h, at 20 μM, in MDA-MB 231 and HepG2 cell lines (Appendix A). These findings are consistent with the findings of Li et al., who hypothesized that the nanoassemblies’ cytotoxicity compared to free DOX is due to the DOX-encapsulated LCNPs’ sustained release properties [20]. However, when the different tumor cell lines were compared, higher cytotoxicity was detected in MDA-MB-231 cells, possibly due to higher sensitivity of this cell line, as confirmed by Gagliardi et al. [64].

DOX-LNLCs were potentially cytotoxic to HepG2 and MDA-MB 231 cell lines after 24 h at concentrations of 1.5 μM or more. On the other hand, DOX-LNLCs were determined to be potentially cytotoxic in normal cell line (HDFn) at concentrations of 3.0 μM or more (Appendix A). Therefore, we can conclude that at 1.5 μM concentration, there was tumor selectivity in HepG2 and MDA-MB 231 cell lines after 24 h. There was no tumor selectivity of the DOX-LNLC in the NCI-H1299 cell line, as the effects were similar to the observed in normal cells. However, it should be noted that the cytotoxicity tests were performed in cells that come into direct contact with the formulations. In the context of in vivo administration, the formulations are first distributed through tissues, and both the triggering effect of acidic pH and the EPR effect, which were not tested in these in vitro cellular assays, could provide tumour selectivity. Moreover, as these tumour cell lines are originated from different tissues (breast, lung, and hepatic), it is possible that variable sensitivities are exhibited as a result of their distinct molecular phenotyping, and thus metabolism when exposed to a specific formulation.

GI_50_ values represent the calculated concentration at which the drug causes 50% inhibition of cell proliferation and are presented, accompanied by the standard deviation (SD) of three independent experiments, in Table 1, as a function of cell line and time. Significantly higher GI_50_ values (*p*-value < 0.05) were obtained for empty LNLCs, confirming that the unloaded nanoassemblies are biocompatible at selected doses. The lower GI_50_ values for DOX, either free or encapsulated, were obtained at 48 h for all the cell lines under analysis. When specifically analysing DOX-LNLCs in comparison to free DOX, it is possible to observe a significant decrease (*p*-value < 0.01) in the GI_50_ in MDA-MB 231 and a smaller decrease in NCI-H1299 after 48 h and in HepG2 after 24 h (*p*-value < 0.05). Therefore, encapsulation of DOX in LNLCs increased its therapeutic potential in comparison to free DOX, and thus a reduction in off-target cytotoxicity risk, which is consistent with those reported in the literature [24,59,65,66] and reinforced by the higher GI_50_ values observed in the normal cell line. 

To elucidate whether the selected LNLCs were efficiently internalized to deliver their drug payload, cell internalization experiments were carried out on the HepG2 tumour cell line and imaged using confocal microscopy. As shown in Figure 10, DOX-LNLCs was mostly internalized and accumulated in the cytoplasm of HepG2 cells at concentrations of 0.05 and 5.0 μM. 

On the other hand, it was observed that at 10 μM concentration, the nanoassemblies tended to accumulate in the cells’ nucleus, and that at this same concentration, a great number of cells were apparently in apoptosis when compared to DOX-LNLCs with lower DOX concentration. Having said that, we can conclude that the intracellular concentration of DOX-LNLCs (10 μM) corresponded well with the limited impact on survival rate observed in the in vitro cytotoxicity assay (Table 1). 

## 4. Conclusions

Our study proposes a rational development of LNLCs preparation and loading methods in terms of colloidal stability. We detected no discernible differences between the methods of producing LNLCs, in contrast to other authors who claim that top-down methods produce more stable cubosome dispersions. We hypothesize that the differences found in the literature may be more influenced by the lipid:polymer ratio than by the method of production. In general, the direct mixing method with the highest EE% and DL% was determined to be the most efficient one when compared to the other drug-= loading methods. Furthermore, when all physicochemical properties were considered, it was determined that DOX-LNLCs M3 (emulsification method) presented a small size (217.4 ± 11.4 nm), an acceptable PDI consistent with a single population type (PDI < 0.25), a high encapsulation efficiency (>90%), a high drug loading content (>7%), and colloidal stability for at least 4 weeks. Furthermore, colloidal stability conferred by polymeric coating compensates for the larger particle sizes in a manner similar to PEGylation strategies. Due to minimal protein binding on the surface of the nanoassemblies, the polymeric coating is also crucial for in vivo serum stability as it may help avoid detection by the immune system’s macrophages and phagocytic system, increase circulation time, and take advantage of the EPR effect. Future assays that are being planned for in vivo studies will confirm the serum stability of these LNLCs nanoassemblies.

Regarding the potential therapeutic performance, the LNLCs have shown a sustained and significantly (*p*-value < 0.05) higher DOX release in a simulated endosomal environment (pH 5.5) when compared to a physiological/plasma environment (pH 7.4). These findings permitted to classify the LNLCs developed as pH-responsive nanoassemblies as the higher drug release triggered by pH decrease being, in turn, a good predictor for achieving successful EPR effect. In addition, the findings indicate the potential of these formulations to minimize cytotoxicity in healthy cells. However, effective selectivity of LNLCs to cancer cells must be confirmed in vivo. Furthermore, encapsulation of the chemotherapeutic agent in LNLCs resulted in a greater decrease in cell growth when compared to DOX free treatment in different tumour cell lines at concentrations of 20 and 45 μM after 48 h and 20 μM after 24 h in HepG2 and MDA-MB 231 lines. The GI_50_ values support these findings, indicating that encapsulating DOX in LNLCs increased its therapeutic potential over the free form. Cellular internalization of the DOX-LNLCs confirmed citotoxicity studies, demonstrating the ability of these hybrid nanoassemblies to increase drug cellular uptake. Furthermore, at higher concentrations, apoptosis and nuclear accumulation of DOX-LNLCs were observed.

## Figures and Tables

**Figure 1 pharmaceutics-15-00326-f001:**
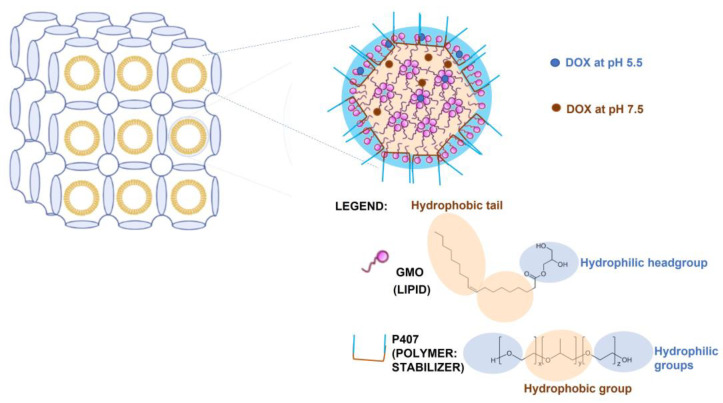
Schematic illustration of LNLCs loaded with DOX. LNLC continuous lipophilic bilayers are shown in yellow and hydrophilic channels in blue, with the amphiphilic block copolymer P407 stabilizer coating the nanoassembly. The drug release is pH-triggered since DOX is incorporated into the lipid matrix of the LNLCs at physiological pH (7.5), but it locates in water regions at acidic pH (5.5) characteristic of endosomal microenvironment.

**Figure 2 pharmaceutics-15-00326-f002:**
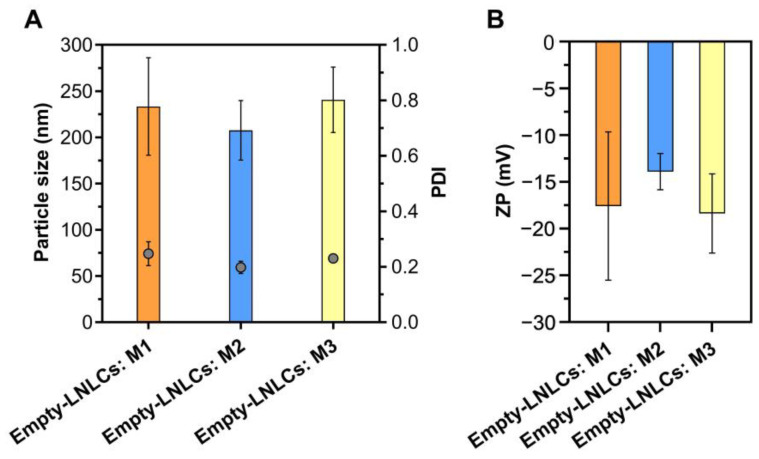
Physicochemical properties of empty LNLCs prepared by hydrotrope method (M1), lipid film hydration method (M2) and emulsification method (M3): (**A**) size and polydispersity index (PDI), and (**B**) surface charge (zeta potential, ZP). The data are expressed as mean ± standard deviation of three independent measurements.

**Figure 3 pharmaceutics-15-00326-f003:**
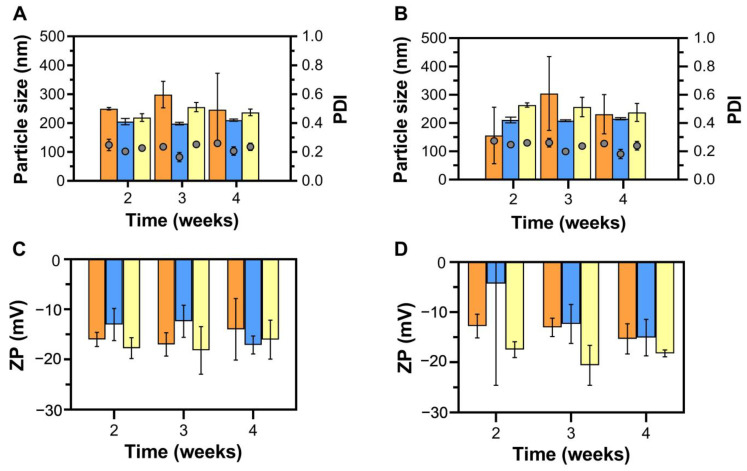
Stability of empty LNLCs prepared by hydrotrope method (M1, orange data), lipid film hydration method (M2, blue data) and emulsification method (M3, yellow data) during 4 weeks in different storage conditions: (**A**) size and PDI at 25 °C, (**B**) size and PDI at 4 °C, (**C**) surface charge (ZP) at 25 °C, and (**D**) surface charge (ZP) at 4 °C. The data are expressed as mean ± standard deviation of least three independent experiments.

**Figure 4 pharmaceutics-15-00326-f004:**
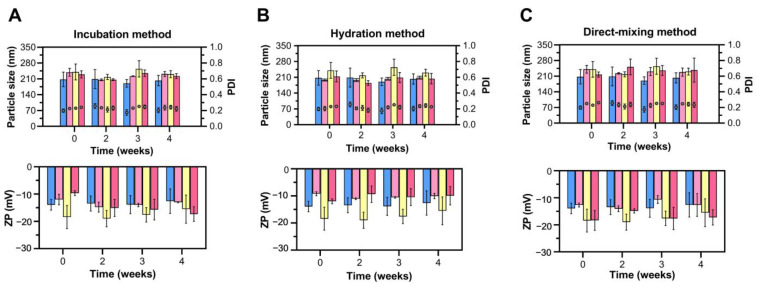
Stability of empty LNLCs and DOX-LNLCs using different loading methods during 4 weeks of storage at 25 °C: (**A**) incubation method, (**B**) hydration method, and (**C**) direct mixing method. Blue data are the empty LNLCS obtained by the lipid film hydration method (M2); yellow data are the empty LNLCS obtained by the emulsification method (M3); and pink data represents the DOX-LNLCs (LNLCs prepared by M2 are lighter pink, while M3 is darker). The data are expressed as mean ± standard deviation of at least three experiments.

**Figure 5 pharmaceutics-15-00326-f005:**
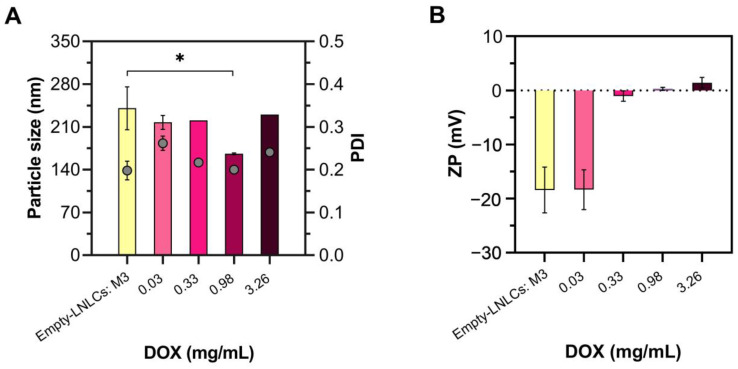
Physicochemical properties of empty LNLCs and DOX-LNLCs with different DOX concentrations (0.03–3.26 mg/mL): (**A**) size and PDI, and (**B**) surface charge (zeta potential, ZP). Yellow data are the empty LNLCS obtained by the emulsification method (M3); pink data represents the DOX-LNLCs (the colour progression (lightest to darkest) represents the increase in DOX concentration (0.03–3.26 mg/mL)). The data are expressed as mean ± standard deviation of three independent measurements. * *p* value < 0.05. One-way ANOVA followed by the Bonferroni post hoc comparative test were performed.

**Figure 6 pharmaceutics-15-00326-f006:**
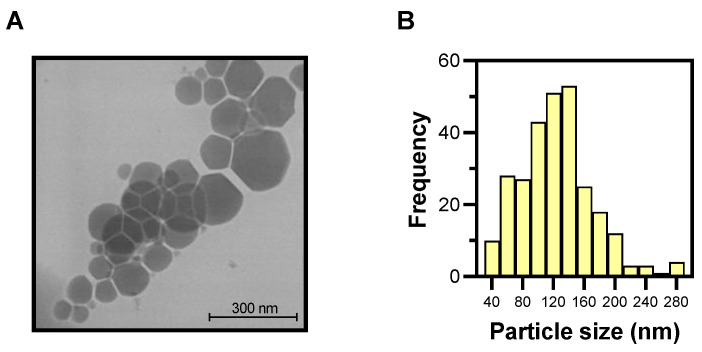
(**A**) STEM photograph of empty LNLCs M3 at 200 kV × 80.00 k. (**B**) Size distribution histogram of empty LNLCs M3.

**Figure 7 pharmaceutics-15-00326-f007:**
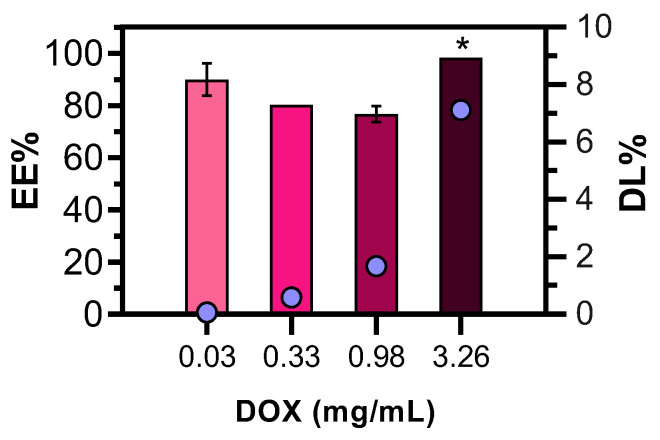
EE and DL of DOX-M3 at different DOX concentrations (0.03–3.26 mg/mL). The data are expressed as mean ± standard deviation of three independent measurements. * *p* value < 0.05. One-way ANOVA followed by the Bonferroni post hoc comparative test were performed.

**Figure 8 pharmaceutics-15-00326-f008:**
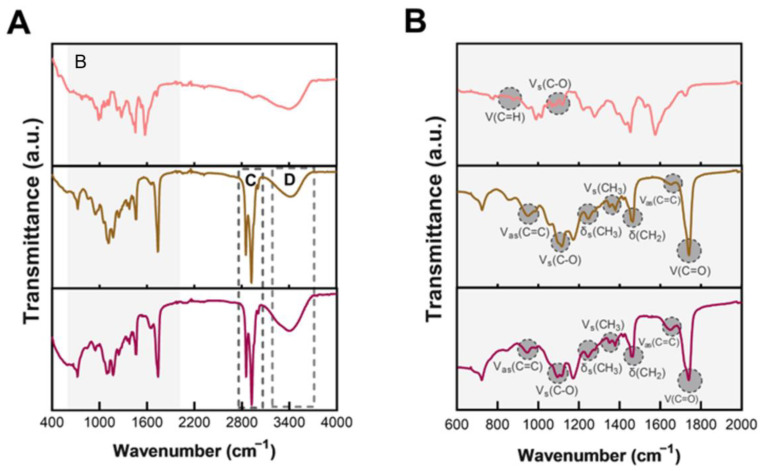
ATR-FTIR spectra of free DOX (pink data—upper spectrum), empty LNLCs (dark yellow data—middle spectrum) and DOX-LNLCs (magenta data—lower spectrum). The shaded area highlights the characteristic functional groups (**A**). Regions B, C (typical vibrations of the chemical groups Vs(CH_2_) at 2860 cm^−1^ and Vas(CH_2_) at 2930 cm^−1^ present in this type of nanoassemblies), and D (v(OH) band between 3200–3400 cm^−1^ assigned to the O-H stretching mode of water, GMO, and P407) are assigned to the regions where functional groups are highlighted. (**B**) Maximization of shaded area (band B). V = stretching vibrations (bond vibrations); δ = deformation vibrations (bending vibrations); s = symmetric; as = antisymmetric.

**Figure 9 pharmaceutics-15-00326-f009:**
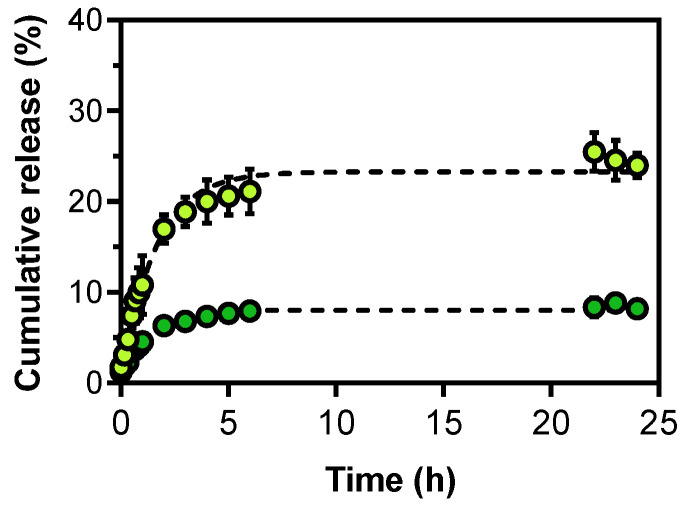
Cumulative DOX release profile from the LNLCs M3 at pH 5.5 (stimulated endosomal environment) (light green data) and pH 7.5 (stimulated physiological/plasma environment) (dark green data) as a function of time (25 h). The data are expressed as mean ± standard deviation of three independent measurements. The lines represent the best kinetic fit obtained according to the first-order model.

**Figure 10 pharmaceutics-15-00326-f010:**
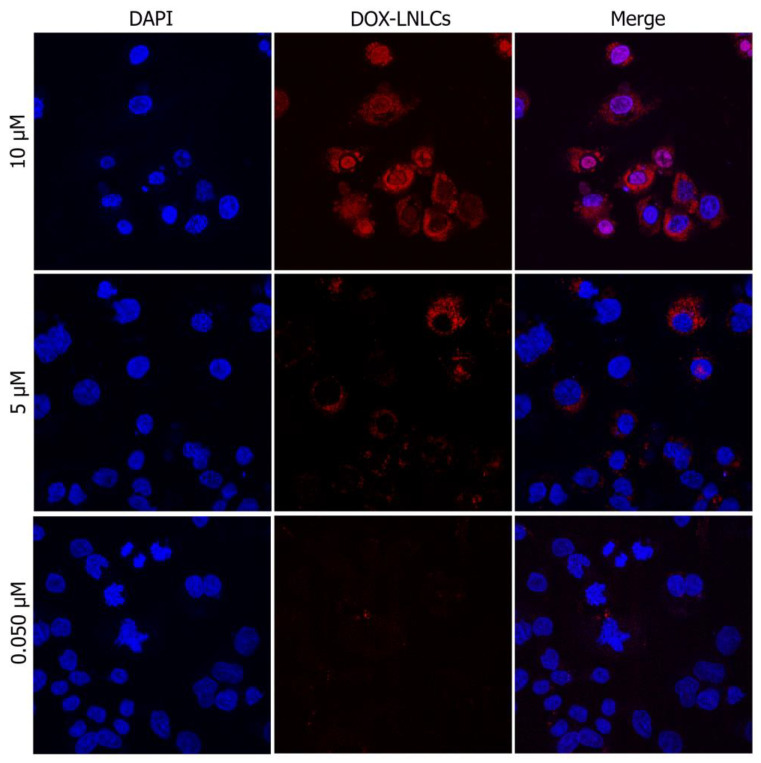
Representative confocal images of cellular internalization and subcellular distribution of DOX-LNLCs nanoassembly in HepG2 cells (63×), after 24 h incubation with DOX concentrations of 10, 5 and 0.05 μM. Cells’ nuclei are counterstained with DAPI (blue, left column); DOX-LNLCs appear in red (middle column), as a result of the emission of DOX. The right column depicts the merged images for each DOX concentration.

**Table 1 pharmaceutics-15-00326-t001:** GI_50_ values and respective deviation standard (SD) of three independent experiments for free DOX, DOX-LNLCs and empty LNLCs as a function of cell lines and incubation time.

Cell Line	Incubation Time (hours)	GI_50_ (μM)		
		Free DOX	DOX-LNLCs	Empty LNLCs
HDFn	24	3.62 ± 0.79	7.83 ± 0.26	36.47 ± 2.64
	48	1.73 ± 0.40	1.03 ± 0.26	17.55 ± 1.63
MDA-MB 231	24	19.43 ± 0.85 1.69 ± 0.25	3.48 ± 0.17	12.51 ± 1.36
	48		1.90 ± 0.78	16.55 ± 1.55
HepG2	24	6.84 ± 1.34	6.48 ± 1.22	19.48 ± 1.35
	48	0.43 ± 0.09	1.05 ± 0.10	22.99 ± 2.09
NCI-H1299	24	6.81 ± 0.71	20.74 ± 2.23	41.98 ± 3.9
	48	7.30 ± 1.79	5.25 ± 0.95	51.50 ± 2.67

## Data Availability

Not applicable.

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
