# Peer review of "pH-Responsive Hybrid Nanoassemblies for Cancer Treatment: Formulation Development, Optimization, and In Vitro Therapeutic Performance"

_pharmaceutics, 2023, doi:10.3390/pharmaceutics15020326_

Round 1

Reviewer 1 Report

The manuscript describes a systematic study of preparation and characterization of nanoassemblies (lyotropic non-lamellar liquid crystaline nanoassemblies) loaded with doxorubicin. The nanoassemblies were designed to release doxorubicin in a pH-dependent manner, and their killing effect in vitro on several cancer cell lines was evaluated. 

General comment: the authors tend to expand - both in the introduction and in the results sections. The suggestion of this reviewer is for the authors to try to condense their text.

The release of doxorubicin is evaluated at pH 5.5 which is characterized by the authors as "the stimulated/characteristic  tumor microenvironment". it is not clear what is meant by this. possibly the authors mean the endosomal environment, in which case they should just state it.

In results section: all three preparation and loading methods result in, more or less, identical formulations in terms of size, zeta potential and specific drug loading. Maybe the authors can compact the text and focus only on the differences (if any).

The particle sizes are relatively large (approximately 220nm ) for the standards of "nanomedicine". Particles of such sizes (especially the ones with -20mV zeta potential) may have significantly short blood circulation times compared to the "standard" 50-100 nm diameter nanoparticles. -- The authors should comment on this and possibly identify routes of administration that may benefit from this particle size. 

On Figure 10: the release kinetics of doxorubicin from the nanocarriers are compared at pH 7.4 and 5.5.  In essence, the difference/advantage of the studied nanocarriers gives an up-to-15% difference in doxorubicin release at acidic conditions compared to the neutral conditions. The kinetics are also of the order of 1-3 hours (for 50% of max release) which is too slow compared to the kinetics of the acidification of the endosomes (pH 5.5 -- it is the understanding of this reviewer that the authors refer to the endosomal pH in this study -- else, the tumor interstitial pH is significantly less acidic -- up to 6.7 -6.8 pH units). However, the kinetics of the endosomal acidification is only of the order of several minutes (maybe up to half hour). -- The authors need to commend on this.

Regarding the data on Table 1 - which lists the GI50 values of the free doxorubicin, doxorubicin in the nanocarriers, and the effect of the empty nanocarriers - there is no information on the number of repeated independent experiments, the mean values and the standard deviations of the reported GI50 values, and the p-values. The statements on lines 680-683 in particular, and the paragraph describing the data on Table 1, lines 672 - 688 should be carefully rewritten  -- some of the statements are not supported by the numbers in Table 1, and statistics are necessary to make any conclusions.

Reviewer 2 Report

The manuscript is to develop a pH responsive hybrid nanoassemblies for cancer treatment. The results looks good, but there are many issues in the manuscript. Also, the manuscript was written with a number of mistakes.

Title

It is prefer to change into, pH responsive hybrid nanoassemblies for cancer treatment: formulation development, optimization, and in vitro therapeutic performance

Abstract

Results and clear conclusion should add to the abstract with minimizing the introduction. Additionally, the abstract intro should not appear as a repeated sentence from the introduction.

Introduction

In general, it was poor and did not give a clear information about the work. I think it should rewrite again. For some example to be added:

Lines 58-60 seems a repeated sentence from the previous paragraph

Lines 66-70 more details about the pH effect on Dox cytotoxicity and cancer cells targeting should added clearly.

Line 83 what meant by colloidal stability

Lines 85- 88 explain what is the relation between polymeric coating and safety

Add a clear explanation in the introduction

Figure 1 no need in the introduction especially its about the final product so you can it in another position

Materials and methods

Abbreviations should add in a certain list or explain each before writing

Al the manuscript should be educed in a rationalize size as the paper is too long

What is the ethanol volume used and the total volume

Line 13 the evaporation clear condition should explained clearly

Line 14 What is the polymer used

Line 21 Explain the DOX solubility in water

Why λ appear in different values add a suitable references to insure its values

Result and discussion

Why the authors consider the preparation method is a preformulation study

Line 36 what is the change done in pH and temperature during preparation according to it the discussion was built on

You should explain the benefit of using three different method. According to the discussion, it was away from the main project target ( effect of pH change).

The discussion appear as a literature review, I was too long , adding no new information

Line 36 what is the change in the pH and Temp done during the preparation explain more details clearly

Line 43-Sentences from 6-9 are not clear

Line 44 add the statistical results

Combine figure S4 with figures F9-A and Fig F9-B

Why using different tiles in the discussion than the method

Conclusion

Conclusion is very poor and should be review again

Round 2

Reviewer 2 Report

The authors succeeded in correcting all the author reviewing commends